# Cyclophilin D, Somehow a Master Regulator of Mitochondrial Function

**DOI:** 10.3390/biom8040176

**Published:** 2018-12-14

**Authors:** George A. Porter, Gisela Beutner

**Affiliations:** Department of Pediatrics, Division of Cardiology, University of Rochester School of Medicine, Rochester, NY 14642, USA; gisela_beutner@urmc.rochester.edu

**Keywords:** cyclophilin D, mitochondrial permeability transition pore, electron transport chain, mitochondrial function

## Abstract

Cyclophilin D (CyPD) is an important mitochondrial chaperone protein whose mechanism of action remains a mystery. It is well known for regulating mitochondrial function and coupling of the electron transport chain and ATP synthesis by controlling the mitochondrial permeability transition pore (PTP), but more recent evidence suggests that it may regulate electron transport chain activity. Given its identification as a peptidyl-prolyl, *cis*-*trans* isomerase (PPIase), CyPD, is thought to be involved in mitochondrial protein folding, but very few reports demonstrate the presence of this activity. By contrast, CyPD may also perform a scaffolding function, as it binds to a number of important proteins in the mitochondrial matrix and inner mitochondrial membrane. From a clinical perspective, inhibiting CyPD to inhibit PTP opening protects against ischemia–reperfusion injury, making modulation of CyPD activity a potentially important therapeutic goal, but the lack of knowledge about the mechanisms of CyPD’s actions remains problematic for such therapies. Thus, the important yet enigmatic nature of CyPD somehow makes it a master regulator, yet a troublemaker, for mitochondrial function.

## 1. Introduction

Cyclophilin D (CyPD, see Table 1 for list of abbreviations) is a member of the cyclophilin family of peptidyl-prolyl, *cis*-*trans* isomerases (PPIases) that resides in the mitochondrial matrix. Despite it first being reported in 1990 [1,2], the exact mechanisms by which CyPD regulates and is regulated by mitochondrial function, remain enigmatic. Cyclophilin D is the product of the *Ppif* gene and has a number of aliases, including mitochondrial cyclophilin, cyclophilin F, cyclophilin 3, and cyclophilin 20. It should not be confused with cytoplasmic cyclophilin D, the product of the *Ppid* gene. Cyclophilin D is an ~22 kDa protein with a mitochondrial targeting sequence that is cleaved as it is imported into the mitochondrial matrix, creating an ~19 kDa final product.

Traditionally, CyPD has been thought to regulate the opening of the permeability transition pore (PTP), a large conductance pore whose opening depolarizes the inner mitochondrial membrane (IMM) to decrease ATP and increase reactive oxygen species production, and drugs that inhibit CyPD have been used in animal and human studies to protect tissues against various insults, such as ischemia–reperfusion injury (IRI). The presence of a number of CyPD binding partners associated with ATP synthase (complex V), such as the adenine nucleotide translocator (ANT), phosphate carrier (PiC), and oligomycin sensitivity conferral protein (OSCP), support the idea that CyPD modulates the PTP through ATP synthase, but this is controversial (Figure 1a). In addition, CyPD may control assembly of the electron transport chain (ETC), making CyPD a central node for control of mitochondrial function (Figure 1b). How CyPD’s PPIase activity is involved in this process remains unclear, as there are very few reports on the mitochondrial targets of this activity (Figure 1c). Finally, recent evidence suggests that the interaction of CyPD with other proteins may also be important for regulating mitochondrial function (Figure 1d). This review builds upon excellent previous reviews of this molecule [3,4,5,6,7] and will discuss these aspects of CyPD and present a model of known CyPD roles in the mitochondrial matrix.

## 2. Mitochondrial Function and the Mitochondrial Permeability Transition Pore

Mitochondria are traditionally known as the powerhouses of the cell, creating energy in the form of ATP through the process of oxidative phosphorylation (OXPHOS). Reducing equivalents from glycolysis, fatty acid oxidation, other substrates, and the Krebs cycle enter complexes I (NADH-ubiquinone oxidoreductase/dehydrogenase) and II (succinate oxidoreductase/ dehydrogenase, also part of the Krebs cycle). Electrons are transferred successively to ubiquinone, complex III (ubiquinol-cytochrome *c* oxidoreductase/dehydrogenase), cytochrome *c*, and complex IV (cytochrome *c* oxidase), which transfers the electron to oxygen, the final electron acceptor (Figure 1b). Driven by these redox reactions, complexes I, III, and IV pump protons from the mitochondrial matrix to the intermembrane space, creating an electrochemical gradient, called the proton motive force (composed of a membrane potential and a proton gradient), across the IMM. ATP synthase uses the energy of the proton motive force to create ATP from ADP and inorganic phosphate. Presumably to increase the efficiency of OXPHOS, the ETC can dynamically form supercomplexes: complexes I, III, and IV can combine with ubiquinone and cytochrome *c* to form respirasomes that may increase the efficiency of proton motive force generation, while ATP synthase can oligomerize with ANT and PiC to form synthasomes that increase the efficiency of ATP production and translocation into the cytoplasm (Figure 1b) [10,11,12,13,14,15]. The proton motive force also controls a number of other transporters within the IMM, including ANT and PiC, which are required to provide the substrates for ATP production and its transport into the cytoplasm.

Mitochondrial coupling is defined as a coordinated link between the activity of the ETC to create the proton motive forces and the activity of ATP synthase to produce ATP, and this requires tight control of IMM electrical resistance; uncoupled mitochondrial have low IMM resistance, such that the proton motive force is dissipated, thus preventing ATP production. This coupling can be regulated to control substrate utilization and mitochondrial ATP production. For example, to generate heat, uncoupling proteins in the IMM can increase the activity of complexes I–IV of the ETC, leading to increased utilization of fuels, such as fats and sugars, without significant increase in ATP production [16]. Mitochondrial coupling also regulates cellular physiology and health, as collapse of the electrochemical gradient initiates apoptosis and/or necrosis by various signals [17].

The PTP is a well-known regulator of mitochondrial IMM coupling, and the history of early research on it is reviewed in more detail in [11,17,18,19]. The phenomenon of an IMM permeability transition was discovered in the 1950s. The term “permeability transition” and the biophysical properties of the PTP were described in 1979 by Hunter and Hayworth [20,21,22] and reviewed in [23]. They and others defined it as a relatively unselective and large pore that allows molecules of up to 1.5 kDa to cross the IMM. It is well-established that sustained opening of the PTP stimulates mitophagy and cellular necrosis [24], but data from many laboratories, including ours, suggests the both sustained and transient opening of the PTP can serve a signaling role in the cell, likely through the regulation of oxidative stress [17,23,25,26,27,28]. Despite much work over the years, understanding the physiology and regulation of the PTP has been hampered by the lack of molecular identification of the PTP, and studies of CyPD have played a central role in this search for the molecule or molecules that create, and do not merely regulate, the PTP.

## 3. Cyclophilin D Regulates the Mitochondrial Permeability Transition Pore

A number of entities have been proposed to create the PTP, and many of these models have been partially or fully discounted so that the real identity of the PTP remains controversial (reviewed in [11,18,29]). Many mitochondrial proteins have been proposed to be the PTP, including ANT and PiC in the IMM, mitochondrial creatine kinase (mtCK) in the intramembrane space, and hexokinase (HK), the translocator protein of 18 kDa (TSPO), and the voltage-dependent anion channel (VDAC) in the outer mitochondrial membrane (OMM), but these were eventually eliminated from consideration [15,30,31,32,33,34,35,36]. In addition, it was proposed that unfolded proteins could create the PTP [37] and that polyphosphate chains and polyhydroxybutyrate could participate in PTP formation [38,39].

Cyclophilin D has also been considered a candidate to form the PTP. In the 1980s, mitochondrial structure, respiration, and calcium levels were found to be affected by cyclosporin A (CsA) [40,41,42]. Eventually, the PTP was found to be inhibited by some cyclosporins, a class of immunosuppressant molecules that inhibit cyclophilins [43,44]. In 1990, the mitochondrial target of CsA was described as a PPIase, whose inhibition also inhibits the PTP [1], and this was confirmed and CyPD was identified as the PPIase within a few years [2,45]. Cyclophilin D was found to be an important regulator of the PTP, but, as with many other potential candidates, the deletion of CyPD merely altered this regulation, and did not eliminate PTP activity, thus eliminating CyPD as a candidate for forming the pore of the PTP [46,47,48,49]. Cyclophilin D appears to control opening of the PTP by sensitizing it to calcium, inorganic phosphate, and perhaps reactive oxygen species, but deletion of CyPD did not alter responses to other PTP stimuli, such as adenine nucleotides, membrane depolarization, pH, thiol oxidants, and ubiquinones [46,47,48,49,50].

Cyclophilin D expression is associated with sensitivity to opening of the PTP. In the rat brain, decreasing expression of CyPD is associated with decreasing sensitivity to PTP opening during neuronal differentiation [51]. In agreement with previous work [1,51], we found that CyPD expression levels are similar in mouse heart and liver, but significantly lower in brain, and these expression levels roughly correlate with CyPD PPIase activity [52]. Although we did not correlate CyPD expression or activity to PTP opening in these tissues, others have shown that sensitivity to PTP opening is highest in liver compared to heart and much lower in brain mitochondria [51,53,54].

## 4. Cyclophilin D Regulates Mitochondrial Function

Both the PTP and CyPD are involved in the regulation of mitochondrial function, but determining independent effects of CyPD is difficult. For example, there is a correlation of decreased CyPD expression with levels of respiration in various organs and during development, but this could be due to direct effects on the PTP [51,55]. In addition, in the aging brain, increased CyPD levels are associated with decreased ATP synthase activity, but there is also increased IMM uncoupling, perhaps from PTP activity, that could explain this effect [56]. However, changes in ATP synthase activity could be due to non-PTP mechanisms. For example, in well-coupled (e.g., PTP closed) heart mitochondria, CsA treatment displaced CyPD from the IMM to the matrix, and this led to increased ATP synthase enzymatic rates, while ATP synthase reaction rates were 50% higher in CyPD null compared to wild type mitochondria [57]. An additional study found similar changes in heart and liver ATP synthase activity when CyPD was inhibited or deleted [58].

We recently studied the correlation of CyPD expression and PPIase activity with synthasome assembly in the mouse heart, liver, and brain [52]. In heart mitochondria, we found that increased respiration and decreased CyPD activity and PTP opening were associated with an increase in synthasome assembly. However, neither CyPD expression nor its total or specific PPIase activity correlated with these changes in different tissues. Instead, and similar to the effects of dissociation of CyPD from the IMM on ATP synthase activity [56,57,58], we found that the CyPD’s PPIase activity relative to ATP synthase protein expression correlate with synthasome assembly [52].

The rest of the ETC may also be affected by CyPD. First, CyPD may mask or otherwise interact with the rotenone inhibition site on complex I [59,60]. Second, overexpression of CyPD in HEK293 cell mitochondria increased respiratory activity, particularly the activity of complex III. In addition, this led to increased assembly of supercomplexes containing complexes I, III, and IV, and faster incorporation of complex III into these supercomplexes, while CyPD was found to bind to complex III and supercomplexes containing complex III [61]. Third, we found that increased activity of ETC complexes I and II and the assembly of respirasomes occurred in the mouse mid-embryonic heart at about mouse embryonic day 11.5 [62], and that this was accelerated by deletion of CyPD [63].

However, one must be careful when evaluating the effects of CyPD inhibition on mitochondrial function and PTP activity when using tissues from animals where CyPD is deleted. These mice have a normal lifespan but are obese [64], but we have seen increased variability in apparent embryonic stage although there are normal litter sizes [26]. This chronic “inhibition” of CyPD can activate compensatory mechanisms that disrupt normal cellular physiology. For example, CyPD null mice have altered mitochondrial calcium levels that activate calcium-dependent mitochondrial dehydrogenases and alter the balance of glucose versus fatty acid oxidation in the heart [65] and increase glucose metabolism throughout the animal [66]. In addition, CyPD deletion in the heart has been found to alter the expression of proteins in the Krebs cycle, and branch chain amino acid and pyruvate metabolism [67]. Furthermore, cardiac CyPD deletion alters mitochondrial protein acetylation patterns that may significantly affect cellular metabolism [68]. One might debate which of these effects are primary and which are secondary to CyPD deletion, but these changes must be taken into account when using CyPD null mouse models.

Therefore, although there may be some experimental caveats, CyPD does appear to regulate mitochondrial OXPHOS, perhaps through both proton motive force generation by complexes I–IV and the utilization of this force by ATP synthase to create ATP. These effects may be caused, in part, by the activation of the PTP, but there are also likely to be PTP-independent effects on mitochondria.

## 5. Cyclophilin D’s Binding Partners

Cyclophilin D has been found to bind to an increasing number of classical and non-classical mitochondrial proteins (see Figure 1d and Table 2 for a summary). Early work suggested an interaction between CyPD, ANT, PiC, and VDAC to create the PTP. Numerous laboratories reported that CyPD complexes with ANT and VDAC to create the PTP [1,69,70,71], and the PiC is also found to bind to CyPD with ANT [72]. Cyclophilin D was associated with complexes of ANT and VDAC with or without HK that had PTP-like electrophysiologic properties, while it was not associated with complexes of ANT, VDAC, and mtCK, which did not have these PTP-like properties [73,74]. However, subsequent deletions of ANT, VDAC, and PiC suggested that they regulated, and did not form, the PTP [30,31,32,33,34]. Therefore, although none of these proteins create the PTP, the interaction of CyPD with ANT, PiC, and other associated proteins is a major regulator of the PTP, but the mechanisms of this regulation remain unclear.

Interactions between CyPD and ATP synthase suggest that the PTP may reside within this important complex. Cyclophilin D interacts closely with the ATP synthase subunits b (ATP5B), d (ATP5D), and OSCP, the latter in a pH-sensitive manner [56,57,75,76], but PTP properties remain despite deletions of subunits b and OSCP [77]. Cyclophilin D binding to OSCP may facilitate deacetylation of CyPD by Sirtuin 3 (SIRT3), as this protein also binds to OSCP [78], while this CyPD–OSCP interaction is inhibited by OSCP’s interaction with estrogen receptor beta [79]. Furthermore, ATP synthase can form synthasomes that also contain ANT, PiC, and mtCK [12,15], which were found to contain PTP-like activity, as discussed above, and we recently reported that CyPD binds to these synthasomes [52]. Further linking CyPD to multimeric complexes of ANT and PiC, a report suggested that the IMM protein, spastic paraplegia 7 (SPG7), may link CypD to VDAC in the OMM [80], but these data have not been confirmed and are controversial [81].

Cyclophilin D is required to elicit PTP-like activity in electrophysiologic experiments using mitochondrial inner membranes and purified components of ATP synthase. The Bernardi group found that a CsA-insensitive, PTP-like activity was present in purified ATP synthase dimers, but not monomers, and that the lack of CyPD in these preparations may explain the insensitivity to CsA [76]. With the Jonas laboratory, we showed that purified ATP synthase monomers displayed PTP-like activity that was sensitive to CsA only if purified CyPD was added to the preparation, and this CyPD/CsA-sensitivity disappeared when purified ATP synthase c-subunits were examined, or if peripheral membrane components of ATP synthase were denatured using urea [82]. Additional data from other laboratories support the idea that the PTP is derived from monomers or dimers of ATP synthase [38,76,82,83,84,85], but others have presented data suggesting that this is not the case [77,86]. This controversy is the subject of many recent reviews [28,29,87,88,89,90,91,92,93,94]. 

A few other classic mitochondrial proteins have been shown to bind to CyPD. As discussed in Section 4, both complex I and complex III of the electron transport chain have been shown to bind CyPD [61]. In addition, AMP kinase (AMPK) induces a dissociation of peroxisome proliferator-activated receptor-α (PPARα) from CyPD, thus inhibiting PTP opening [95].

A number of proteins involved in various cellular signaling pathways have been found to interact with CyPD. Cyclophilin D binds to both mitogen-activated protein kinase 1 (ERK) and glycogen synthase kinase-3β (GSK-3β) in the mitochondrial matrix [96,97], and the phosphorylation/deactivation and mitochondrial translocation of GSK-3β decreases the interaction of CyPD and ANT and suppresses PTP opening in a developmentally regulated manner [55,98]. Mammalian sterile 20-like kinase 1 (MST1) lies upstream of protein kinase pathways, and was shown to enter mitochondria and bind to CyPD to enhance cell death [99].

Protective and stress response pathway proteins can also bind to CyPD. In cancer cells, CyPD binds to a complex of tumor necrosis factor type 1 receptor-associated protein (TRAP-1) and its related proteins heat shock protein 90 (HSP60) and heat shock protein 90 (HSP90) in the mitochondrial matrix; this binding sequesters CyPD and prevents it from opening the PTP, while an HSP90 inhibitor releases CyPD and causes PTP opening [100,101,102]. In contrast, the cell-stress response protein, p53, binds to CyPD to induce the PTP in various cell types [103,104]. This important protein may enter the mitochondrial matrix via as many as three redox/respiration dependent import pathways, unfolding as it enters [105,106]. The p53 and TRAP-1 pathways may be linked, as a model in which oxidative stress induces p53 translocation into the mitochondrial matrix where unfolded p53 then binds to and displaces CyPD from TRAP-1, increasing CyPD’s PPIase and PTP-opening activity [105]. Perhaps in a related pathway, overexpression of the DnaJ heat shock protein family (HSP40) member C15 (DNAJC15) increased PTP opening and, in stressed cells, DnaJC15 and heat shock protein 70 (HSP70) were found to bind to CyPD [102].

Cyclophilin D binds to B-cell lymphoma 2 (BCL2) in a CsA-sensitive manner but independent to its binding to ANT and changes in calcium levels. This interaction increases BCL2’s prevention of BH3 Interacting Domain Death Agonist (tBID)-mediated cytochrome *c* release from mitochondria, an initiating event in apoptosis that is not related to the PTP [107]. Signal transducer and activator of transcription 3 (STAT3) binds to CyPD and induces cardioprotection, although the mechanisms by which this occurs are unclear [108,109]. Tissue kallikrein also induces cardioprotection, perhaps by association with CyPD and VDAC, as demonstrated by proximity ligation assays [110].

Finally, a few other proteins that have been shown to modify PTP activity can bind to CyPD. Complement 1q-binding protein (C1QBP), which may inhibit the PTP, may protect cells from injury by binding to CyPD, but not ANT or VDAC, in the mitochondrial matrix [111]. The enzyme 2’,3’-cyclic nucleotide 3’-phosphodiesterase (CNP) can bind to CyPD, perhaps indirectly via binding to ANT, to increase PTP opening [112].

## 6. Cyclophilin D’s Enzymatic Activity

Cyclophilin D was initially identified by the ability of CsA to inhibit its PPIase activity [1,2,40,44], and a working model of the molecular mechanism of CyPD’s PPIase activity based on its similarity to cyclophilin A was recently presented [6]. Therefore, the general consensus is that this PPIase activity is important for CyPD’s control of mitochondrial function. However, there is very little data confirming that CyPD participates in protein folding. Some reports demonstrated that the mitochondrial cyclophilins of yeast/molds participate in mitochondrial protein folding upon importation in cooperation with HSP60 and HSP70 [113,114]. One manuscript showed that CyPD’s isomerization from *trans* to *cis* of proline residue 61 maintains rat liver ANT in its cytoplasmic (“c”) facing conformation to cause opening of the PTP [115], while a study of mouse embryonic fibroblasts and a human cell line found that CyPD may isomerize unfolded p53 to create fibrils in the mitochondrial matrix [105].

Cyclophilin D’s PPIase activity has been shown to protect cells from apoptotic stimuli, but this protective effect of CyPD on cellular viability is not widely reported outside of cancer cells and may be unrelated to CyPD’s control of the PTP [107,116]. In addition, re-expression of an isomerase-inactive CyPD did not rescue the wildtype phenotype in CyPD null fibroblasts [46]. However, it has also been suggested that CyPD’s control of the PTP may be independent of its PPIase activity [117], as has been found for the inhibition of calcineurin by cyclophilin A [118,119]. In addition, ablation of CyPD’s PPIase activity and/or substrate binding does not affect its ability to bind to ANT, an important mechanism in PTP regulation [116], although ANT is reported to be a target of CyPD’s PPIase activity [115].

We recently suggested that CyPD may control ATP synthase-containing synthasome assembly, but we did not find that the expression levels or absolute or specific activity of CyPD correlated with synthasome assembly [52]. Therefore, CyPD’s regulation of synthasome assembly may not be related to CyPD’s PPIase activity. However, CyPD’s PPIase activity relative to ATP synthase protein expression was associated with changes in synthasome assembly [52]. These findings may be important for the PTP, as we presented a model in which increased synthasome assembly decreases the probability of PTP formation from ATP synthase monomers or dimers. If this model is correct, then it remains undetermined whether and how CyPD PPIase activity controls this process.

The standard assay for measuring PPIase activity was first reported by the Fischer group in the early 1980 [120,121] and has been used in many publications since with minor alterations [122,123]. In addition, a newer, fluorescence-based assay was recently reported [105]. We have employed the older assay [52] and find that, to discern the specific activity of mitochondrial CyPD, purified mitochondria must be used to eliminate the measurement of non-mitochondrial PPIases. In addition, it is important to include controls containing CsA, as other PPIases are present in mitochondria. One should also avoid the use of protease inhibitors during processing of the tissue, as we found that phenylmethylsulfonyl fluoride (PMSF) and commercial protease inhibitor cocktails inhibit PPIase activity but not the ability of CyPD to bind or remain bound to its target protein [63]. Finally, using a spectrophotometer with fast reading rates make is easier to measure the rapid upstroke of absorbance that occurs when the reaction is initiated [63].

## 7. Physiologic Regulation of Cyclophilin D

Most early experimental dissection of CyPD’s function involved its deletion, inhibition, or overexpression, but more recent work has defined the physiologic regulation of CyPD by post-translational modifications, including acetylation, oxidation, *S*-nitrosation, *S*-glutathionylation, and phosphorylation. A better understanding of the mechanisms by which CyPD undergoes these post-translational modifications could provide therapeutic targets to modulate CyPD activity in patients.

### 7.1. Acetylation

Acetylation is an important post-translational modification that responds to the metabolic state of the cell. Spontaneous acetylation can occur in conditions observed in the mitochondrial matrix (an alkaline environment and abundance of acetyl-CoA [124]), while one mitochondrial transacetylase has been described [125,126,127]. Acetylation of cyclophilin A at lysine 125 inhibits its PPIase activity and cyclosporin binding, leading to increased calcineurin activity [128]. Cyclophilin D can be acetylated at lysine residue 166, which is close to the PPIase catalytic site, and this residue can be deacetylated by SIRT3, the major deacetylase in mitochondria [129]. Two manuscripts showed that acetylation of this residue increases CyPD’s PPIase activity but, unfortunately, they were retracted, and the data supporting this finding was difficult to interpret. Since then, no studies have been published that confirm this finding.

Other circumstantial evidence suggests that CyPD acetylation is an important physiologic mechanism. Inhibition of this deacetylation using SIRT3 deletion caused increased death when mice were subject to stress by transaortic constriction [129], while heart failure decreased the expression of SIRT3 and increased CyPD acetylation [130]. An additional report demonstrated that hypoxia/ischemia increased CyPD acetylation but that overexpression of SIRT3 attenuated this acetylation and the resulting pathology in vitro. Furthermore, expressing a CyPD acetylation mimic decreased calcium retention and increased cell death while expression of a deacetylated CyPD mimic had the opposite effects in this cardiac cell line. In vivo, IRI increased CyPD acetylation, while ischemic conditioning, which decreased injury, decreased this acetylation in the heart, presumably via SIRT3, as these effects were lost in SIRT3 null mice [131]. Similar effects of IRI in the heart and brain were rescued by hypothermia and CyPD inhibition [132]. Finally, IRI in liver increased CyPD acetylation levels and treatment with NAD^+^, a SIRT3 substrate, ameliorated this effect [133].

### 7.2. Oxidation/Nitrosation/Glutathionylation

As with acetylation, oxidation is an important post-translational modification in mitochondria, as much of the cellular reactive oxygen species is produced by the ETC. The cysteine residue 203 (human sequence) of CyPD is redox-sensitive, and its oxidation does not alter PPIase activity [134]. This residue was found to undergo *S*-nitrosation in vivo [135], and treatment of mouse embryonic fibroblasts with a nitric oxide donor protected from PTP opening induced by oxidative stress [136]. In addition, expression of CyPD that was mutated to prevent modification of this reside (C203S) also protected cells from PTP opening in vivo and in vitro [136]. Furthermore, tachycardia preconditioning decreased PTP opening and increased *S*-glutathionylation of CyPD [137]. These results suggest that these modifications prevent oxidation of C203S residue, a deleterious modification that could help activate the PTP, although it should be noted that *Bos taurus* CyPD lacks this particular residue, yet displays a canonical CsA-sensitive PTP.

### 7.3. Phosphorylation

The importance of phosphorylation in the regulation of mitochondrial proteins has been questioned; although kinases have been shown in mitochondrial preparations, it is unclear how they translocate into the mitochondrial matrix from the cytoplasm without an obvious mitochondrial targeting sequence, and there are no known phosphatases to counteract their activity. However, there are some reports that suggest that CyPD can be phosphorylated to enhance its activity. Cyclophilin D can be phosphorylated by GSK-3β, and this is associated with increased PTP opening in a cell line [96]. In a rat liver model, increased CyPD phosphorylation was induced after IRI was decreased after treatment with an inhibitor of GSK-3β [133]. The serine/threonine kinase 2 was also shown to phosphorylate CyPD at serine residue 31, and CyPD null cells expressing a dephosphorylation mimic (S31A) lacked PPIase activity and exhibited disruption of multiple bioenergetic pathways and, in contrast to the above, evidence of PTP opening [138]. Finally, it was reported that deletion of the calcium uniporter increased CyPD phosphorylation at S42, CyPD’s association with ATP synthase, and sensitivity of the PTP to calcium [139].

## 8. Therapeutic Potential of Targeting Cyclophilin D

Since the PTP is an important mediator of cell necrosis, a major goal of research over the last 30 years has been to design pharmacologic reagents to prevent opening of the PTP and the subsequent organ pathophysiology. Much of this work has focused on IRI, although models of different pathologic states have been studied. From the beginning, this work has focused on inhibiting CyPD using CsA, its analogues, and unrelated reagents.

The beneficial effects of CyPD inhibition on IRI have been reported in too many manuscripts to be addressed completely in this review, but a few notable examples will be discussed. Cyclosporin A was first found to be protective in IRI in vivo in 1993 [140]. Inhibition or deletion of CyPD has been used to show a similar protection from IRI in the heart [141], skeletal muscle [142], lung [143], brain [144], and kidney [145]. Regarding the heart, the Ovize laboratory translated this idea into the clinic by treating patients with acute myocardial ischemia with CsA during percutaneous coronary intervention. Although initial studies suggested that this therapy was somewhat effective [3,146], a larger study failed to show substantial benefit from this therapy [147], perhaps due to the formulation of CsA used [148,149].

Further work must be done to determine if these treatments can be used clinically in the ischemic heart and in other models of human disease. For example, we recently published data suggesting that inhibition of CyPD may increase cardiac function in the neonatal heart [27]. Cyclophilin D inhibition using CsA or its analogue, NIM811, increased mitochondrial maturity and myocyte differentiation in cultured neonatal myocytes. Treatment of neonatal mice with daily injections of these reagents significantly increased cardiac function at 7 days of age. Cyclophilin D null mice had similar findings both in vivo and in vitro [27]. Therefore, it is possible that CyPD inhibition could be used to increase cardiac function in neonates who have cardiomyopathy due to a variety of conditions.

For these treatments to be effective, appropriate drugs must be available. Cyclosporin A is the prototypical CyPD inhibitor, but it also inhibits cyclophilin A and the calcineurin pathway, an effect that has long been used for immunosuppression in humans. A number of non-immunosuppressive analogues of CsA, which do inhibit CyPD, have been developed, including NIM811 (*N*-methyl-4-isoleucine-cyclosporin), Alisporivir (Debio025), and MeVal-4-cyclosporin (*N*-methylvaline-4-cyclosporin A) (Figure 2a) [19,150,151,152]. In addition, mitochondrial-targeted analogues have been developed to increase the relative concentration of CsA in mitochondria (Figure 2b) [153,154,155]. Additional CyPD inhibitors, not related to CsA, include Sanglifehrin A, antamanide, ER-000444793, and urea-derived compounds (Figure 2c) [156,157,158,159,160], while small molecule screens have identified other inhibitors of CyPD [161]. In addition, given the many CyPD binding partners and post-translational modifications, it is possible that inhibitors of pathways that regulate CyPD might also be used to modulate its action. For example, an inhibitor of the TRAP-1/HSP90 complex, Gamitrinib, may also modulate CyPD activity [138].

## 9. Conclusions

Cyclophilin D is a small but complex molecule that appears to play an important role in mitochondrial biology, but the mechanisms by which it does this are still obscure (Figure 1). It is a chaperone protein that can help fold other proteins, but there is scant evidence that it does this in the mitochondrial matrix. Alternatively, CyPD may play a role as a scaffolding protein, as evidence suggests that it binds to many proteins in the matrix, particularly those involved in stress responses. Finally, it undergoes post-translational modifications that likely regulate its activity.

How these aspects of CyPD control and/or participate in its recognized function of PTP regulation remains a mystery. Data is conflicting on the role of PPIase activity in PTP regulation, although it is likely that this enzymatic activity is important for this function. In addition, the multiple binding partners for CyPD suggest a complex regulation of its control of the PTP, as would be expected for such an important cellular phenomenon. Finally, the PPIase and scaffolding activities of CyPD are not mutually exclusive, as protein folding may affect CyPD’s binding to some proteins, e.g., p53, while binding to protein complexes may control whether active CyPD is free to fold proteins in the matrix.

Finally, although CyPD likely has other functions in the mitochondrial matrix, its main function is thought to be regulation of the PTP. First, more work must be done to discover and dissect these non-PTP functions. Experiments must also be done to determine the importance and mechanisms of CyPD’s regulation of synthasome assembly and ETC activity. Finally, until the identity of the PTP is determined, it will remain difficult to fully understand how CyPD controls the PTP, and if this is related to its regulation of the ETC. Thus, despite all of the work demonstrating the importance of CyPD as a master regulator of mitochondrial function, our lack of knowledge of this phenomenon makes it also a troublemaker when trying to understand mitochondrial function.

## 10. Patents

G.A.P.J. has a patent pending entitled “Compositions and methods for enhancing cardiac function in the neonate” that is related to this subject.

## Figures and Tables

**Figure 1 biomolecules-08-00176-f001:**
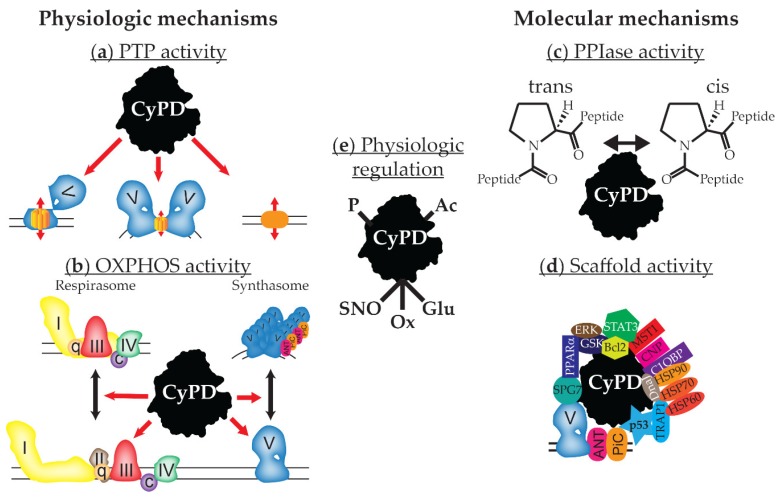
A model of the molecular and physiologic mechanisms of cyclophilin D (CyPD)’s action and regulation. (**a**) The major known physiologic function of CyPD is regulation of the mitochondrial permeability transition pore (PTP). It remains unclear how CyPD regulates the three models of the PTP presented: the c ring of ATP synthase (**left**), dimers of ATP synthase (**middle**), and an unknown entity in the inner mitochondrial membrane (**right**). (**b**) Data also suggests that CyPD may regulate oxidative phosphorylation (OXPHOS) activity, perhaps altering the activity of the respiratory chain and respirasome assembly and inhibiting the activity of ATP synthase and synthasome assembly (electron transport chain (ETC) complexes and ATP synthase are labeled with their complex number, while q and c designate coenzyme q/ubiquinone and cytochrome *c*, respectively. (**c**) CyPD is a peptidyl-prolyl, *cis*-*trans* isomerase (PPIase) that resides in the mitochondrial matrix, but the targets of this PPIase activity are poorly defined. (**d**) CyPD also functions as a scaffold protein, bringing various structural and signaling molecules together to effect changes in mitochondrial physiology. (**e**) CyPD’s activity is regulated by its expression, which is developmentally regulated in some organs, and its post-translational modification, shown as phosphorylation (P), acetylation (Ac), *S*-nitrosation (SNO), oxidation (Ox), and *S*-glutathionylation (Glu).

**Figure 2 biomolecules-08-00176-f002:**
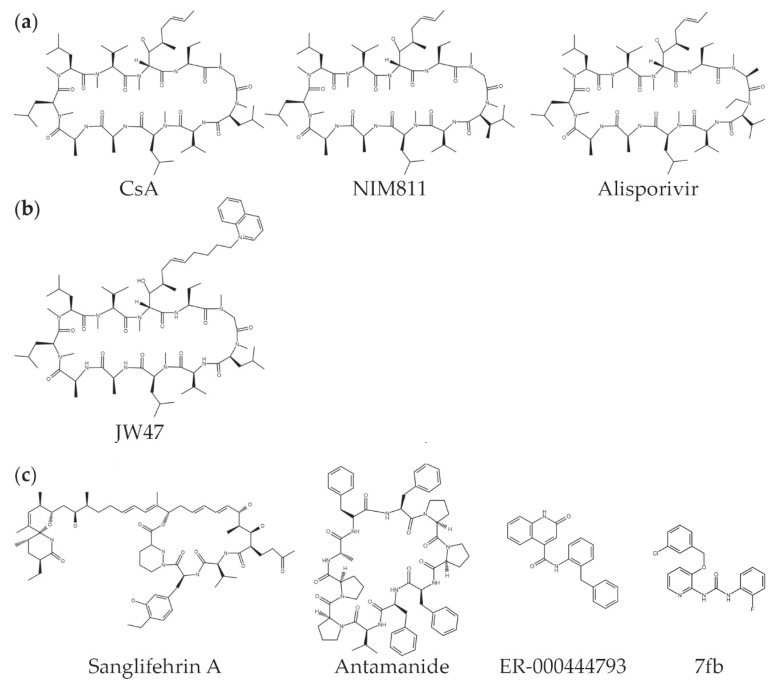
Structures of various CyPD inhibitors. (**a**) Cyclosporin A (CsA) and two of its common derivatives, NIM811 ((Me-Ile-4)cyclosporin A), and Alisporivir (Debio-025, MeAla(3)EtVal(4)-cyclosporin). (**b**) To increase its concentration in the mitochondrial matrix, a positively charged, quinolinium cation was tethered to CsA to create compound JW47 [155]. (**c**) CyPD inhibitors not derived from CsA include Sanglifehrin A [158], antamanide [156], ER-000444793 [157], and compound 7fb (1-(3-(3-chlorobenzyloxy)pyridin-2-yl)-3-(2-fluorophenyl)) [159]. Structures were drawn using the program “Marvin JS” by ChemAxon at the Fischer Scientific website [162] and based on structures in [163] (**a**,**b**) and the references given (**b**,**c**).

**Table 1 biomolecules-08-00176-t001:** List of abbreviations.

Abbreviation	Full Name	Gene Name ^1^
AMPK	AMP kinase	
ANT	adenine nucleotide translocator	*Slc25a4-6*
ATP5PB	ATP synthase peripheral stalk-membrane subunit B	*Atp5pb*
ATP5PD	ATP synthase peripheral stalk-membrane subunit D	*Atp5pd*
BCL2	B-cell lymphoma 2	*Bcl2*
C1QBP	complement 1q-binding protein	*C1qbp*
CNP	2′,3′-Cyclic Nucleotide 3’ phosphodiesterase	*Cnp*
CsA	cyclosporin A	
CyPD	cyclophilin D	*Ppif*
DNAJC15	DnaJ heat shock protein family (Hsp40) member C15	*Dnajc15*
ERK	mitogen-activated protein kinase 1	*Mapk1*
ETC	electron transport chain	
GSK-3β	glycogen synthase kinase-3β	*Gsk3b*
HK	hexokinase	*Hk1-4*
HSP60	heat shock protein 60	*Hsp60*
HSP70	heat shock protein 70	*Hsp70*
HSP90	heat shock protein 90	*Hsp90*
IMM	inner mitochondrial membrane	
IRI	ischemia–reperfusion injury	
kDa	kilodaltons	
MST1	mammalian sterile 20-like kinase 1	*Stk4*
mtCK	mitochondrial creatine kinase	*Ckmt1,2*
OMM	outer mitochondrial membrane	
OSCP	oligomycin sensitivity conferral protein	*Atp5po*
OXPHOS	oxidative phosphorylation	
PiC	mitochondrial phosphate carrier	*Slc25a3*
PMSF	phenylmethylsulfonyl fluoride	
PPARα	peroxisome proliferator-activated receptor-α	*Ppara*
PPIase	peptidyl-prolyl, *cis*-*trans* isomerase	
PTP	mitochondrial permeability transition pore	
SIRT3	sirtuin 3	*Sirt3*
SPG7	spastic paraplegia 7	*Spg7*
STAT3	signal transducer and activator of transcription 3	*Stat3*
TRAP-1	tumor necrosis factor type 1 receptor-associated protein	*Trap1*
tBID	BH3 interacting domain death agonist	*Bid*
TSPO	translocator protein of 18 kDa	*Tspo*
VDAC	voltage-dependent anion channel	*Vdac1-3*

^1^ If applicable, human gene nomenclature is from GeneCards ([8,9]).

**Table 2 biomolecules-08-00176-t002:** CyPD binding partners.

Protein	Binding to CyPD ^1^	References
ANT	Direct	[1,69,70,71]
ATP5PB	Possibly direct	[57,75]
ATP5PD	Possibly direct	[57,75]
BCL2	Direct	[107]
C1QBP	Direct	[111]
CNP	Indirect	[112]
DNAJC15	Direct	[102]
ERK	Direct	[96]
GSK-3β	Direct	[55,96,97]
HK	Indirect	[73,74]
Hsp60	Direct	[100,101]
Hsp70	Direct	[102]
Hsp90	Direct	[100,101]
MST1	Direct	[99]
mtCK	Indirect	[73,74]
OCSP	Direct	[56,57,75]
p53	Direct	[103,104]
PiC	Direct	[72]
PPARα	Direct	[95]
SIRT3	Indirect	[78]
SPG7	Direct	[80]
STAT3	Direct	[108,109]
Tissue kallikrein	Possibly direct	[110]
TRAP-1	Direct	[100,101,102]
VDAC	Indirect	[69,80]

^1^ Binding associations are listed as ‘Direct’ if immunoprecipitation or affinity matrix assays were used, ‘Possibly Direct’ if proximity assays were used, and ‘Indirect’ if there is no clear direct binding but the protein has been physically associated with CyPD.

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
