# Peer review of "Cyclophilin D, Somehow a Master Regulator of Mitochondrial Function"

_biomolecules, 2018, doi:10.3390/biom8040176_

Reviewer 1 Report

This is a timely review which adequately covers past and recent findings on the role of Cyclophilin D in mitochondrial function.  However some textual revisions are necessary, and Authors should carefully check the text for other misprints. The English is not very fluent and the style is often convoluted.

1.       The ATP synthase (complex V) is not a component of the electron transport chain (ETC)!! Please revise lines 40-41 and lines 178-179 which contain this erroneous statement.

2.       Lines  42-43: the sentence “… CyPD controls the creation of the PTP from ATP synthase” needs to be changed to “… CyPD modulates the PTP through ATP synthase” (or a similar statement) because genetic ablation experiments (refs 25-28) have demostrated that the PTP forms and opens even in the absence of CyPD.

3.       Line 61: “inner membrane” should be “intermembrane” space.

4.       Line 94: there is something wrong in the sentence “Mitochondrial proteins that regulate the pore were proposed to be the PTP”.

5.       Lines 186-188: Change “… in patch clamp experiments..” to “…. in electrophysiological experiments ….” because ref 65 reports bilayer experiments, not patch clamp.

6.       Line 257-259, this sentence should be rewritten as it does not stand gramatically.

7.       Lines 273-275: “.... as it requires” should be changed because the statement that follows does not provide an explanation to the previous sentence; the mitochondrial matrix does not have a “low” pH, if this is intended to mean “acidic”, as most measurements indicate that it is more alkaline than the cytosol (which makes sense).

8.       This topic has been reviewed before and it would be useful to mention this earlier coverage by Giorgio et al. (2010) Cyclophilin D in Mitochondrial Pathophysiology, Biochim. Biophys. Acta 1797, 1113-1118 and by Javadov and Kuznetsov (2013) Mitochondrial permeability transition and cell death: the role of cyclophilin D, Front.Physiol. 4. 76.

Author Response

We thank the Editor and Reviewers for the constructive comments of our manuscript. We admit that some mistakes were made as we rushed to finish this manuscript and get it in well past the deadline. We regret these mistakes and have made the requested changes as well as other changes in the text to make the manuscript more readable. After each comment below, we have described the changes we have made, in detail as needed, in bold text. As suggested, we have also added two figures; one with a model of CyPD’s action and another with chemical structures of some CyPD inhibitors. We hope that the changes are acceptable and that this manuscript will be accepted in the near future.

Academic Editor Comments:

“Probably 'trouble maker' in the title is not so good?”
We have removed this from the Title, but kept the uncertainty of CyPD’s activity with the new title: “Cyclophilin D, somehow a master regulator of mitochondrial function.” However, we have kept this term in the Conclusion.

Reviewer 1:

1.       The ATP synthase (complex V) is not a component of the electron transport chain (ETC)!! Please revise lines 40-41 and lines 178-179 which contain this erroneous statement.

Thank you for pointing this out. We have previously stated this in grants and other manuscripts and should have stated it herein, as you suggest. In lines 40-41, we now state “ATP synthase (complex V)” and have removed reference to the ETC here. In line 179, we have replaced “this ETC complex” with “this important complex”.

2.       Lines  42-43: the sentence “… CyPD controls the creation of the PTP from ATP synthase” needs to be changed to “… CyPD modulates the PTP through ATP synthase” (or a similar statement) because genetic ablation experiments (refs 25-28) have demostrated that the PTP forms and opens even in the absence of CyPD.

This change has been made.

3.       Line 61: “inner membrane” should be “intermembrane” space.

This change has been made.

4.       Line 94: there is something wrong in the sentence “Mitochondrial proteins that regulate the pore were proposed to be the PTP”.

We have changed this to “Many mitochondrial proteins have been proposed to be the PTP. . .”

5.       Lines 186-188: Change “… in patch clamp experiments..” to “…. in electrophysiological experiments ….” because ref 65 reports bilayer experiments, not patch clamp.

We have made this change, but note that this sentence has been expanded based on the comment of Reviewer 2.

6.       Line 257-259, this sentence should be rewritten as it does not stand gramatically.

We agree and have rewritten this sentence.

7.       Lines 273-275: “.... as it requires” should be changed because the statement that follows does not provide an explanation to the previous sentence; the mitochondrial matrix does not have a “low” pH, if this is intended to mean “acidic”, as most measurements indicate that it is more alkaline than the cytosol (which makes sense).

We have added a description of how acetylation can occur in mitochondria. The statement of “low pH” was a mistake and we agree that alkaline is a better term.

8.       This topic has been reviewed before and it would be useful to mention this earlier coverage by Giorgio et al. (2010) Cyclophilin D in Mitochondrial Pathophysiology, Biochim. Biophys. Acta 1797, 1113-1118 and by Javadov and Kuznetsov (2013) Mitochondrial permeability transition and cell death: the role of cyclophilin D, Front.Physiol. 4. 76.

We have added reference to these and other previous reviews at the end of the introduction.

Reviewer 2 Report

This review article by Porter and Beutner critically covers some previous and recent developments on cyclophilin D (CypD) and its known roles on mitochondrial function and the mitochondrial permeability transition pore (PTP) as a potential target for pharmacological research.

Of particular relevance is the fact that the authors detail a thorough list of known CypD binding partners. In addition, the manuscript features a well-tailored List of abbreviations table for inexperienced readers and well-ordered sections covering how the PTP governs mitochondrial function and CypD as PTP regulator, chaperone, enzyme and target for regulation in a disease-control context. While this review is well-structured and indeed offers theoretical advance over previously published reviews, I do have some concerns on basic concepts and recommendations for the authors before further considering this manuscript.

Concerns:

1.   Please note that mitochondrial coupling refers to the coordinated link between respiratory chain activity and ATP synthesis. That being said, inner mitochondrial membranes cannot be coupled. Please change this in the Abstract section and make this more clear on line 72.

2.   This review article would be more beneficial if some sort of figure encompassing CypD´s “master regulator” versus “trouble maker” roles are outlined.

3.   Sentence on line 44,45 and 46 sort of contradicts sentence on line 46 and 47.

4.   On line 48 change “…model of CypD activity in the mitochondrial matrix.” For “…model of known CypD roles in the mitochondrial matrix.”

5.   Please note on line 57 that complex II is part of the Krebs cycle.

6.   On line 68, please refer the reader to specific research articles and not only reviews by your group. In other words, change reference #4 for #61. By the way, not everyone has accepted the “Synthasome” concept. For example, note that your Scientific Reports paper (PMID: 29101324) does not show Phosphate Carrier immunodetection alongside ADP/ATP translocase and ATP synthase.

7.   Please cite PMID: 26902508 on line 81. 

8.   On line 84 cite an excellent review on Hunter and Haworth´s work PMID: 9714722.

9.   Add a suitable reference on line 98.

10.On line 85 add a suitable reference after “necrosis”.

11.On line 90 add “componentry” or something alike after PTP.

12. Change ubiquinone for ubiquinones on line 112.

13. The OSCP sentence feels out of context on lines 124-126. Try to merge lines 127-132 in one ATP synthase and CypD paragraph.

14.By the way, the authors are not citing a key paper showing unchanged PTP responsiveness to calcium and cyclosporin A in OSCP depleted cells (PMID: 28784775).

15. Add cyclosporin A (CsA) to the list of abbreviations and update the manuscript accordingly.

16. Sentence on lines 173-175 is misleading considering that HK and mtCK have not been deleted and PTP under such conditions assessed.

17.For the sake of balance, please cite and explain the findings in PMID: 28289229.

18. On line 180 add a sentence something like “Although these results have been seriously challenged [PMID: 28784775]”

19.On line 186 add a similar sentence and cite PMID: 26581158.

20. On line 188 change “PTP activity” for “CsA-insensitive, PTP-like activity”.

21.Please discuss a bit on how can p53 enter the mitochondria on line 205.

22. Add a citation after “synthasome assembly” on line 249.

23. On line 254, please note that Molkentin´s group has previously shown that prolyl isomerase-deficient CypD recapitulates the CypD knockout phenotype suggesting CypD enzymatic activity is indeed required for PTP (PMID: 15800627).

24. On section 8.2, please note that Bos Taurus CypD lacks an equivalent C203 and yet displays a canonical CsA-sensitive PTP.

25. On line 307 change “…without a mitochondrial…” for “…without an obvious mitochondrial…” and “no phosphatases” for “not known phosphatases”.

26.On line 331, please mention possible causes for this outcome (check Bernardi´s Letter to the Editor after reference 115).

27.Why don´t you add a figure showing the molecular structure/family of all known drugs targeting CypD.

Author Response

We thank the Editor and Reviewers for the constructive comments of our manuscript. We admit that some mistakes were made as we rushed to finish this manuscript and get it in well past the deadline. We regret these mistakes and have made the requested changes as well as other changes in the text to make the manuscript more readable. After each comment below, we have described the changes we have made, in detail as needed, in bold text. As suggested, we have also added two figures; one with a model of CyPD’s action and another with chemical structures of some CyPD inhibitors. We hope that the changes are acceptable and that this manuscript will be accepted in the near future.

Reviewer 2:

1.   Please note that mitochondrial coupling refers to the coordinated link between respiratory chain activity and ATP synthesis. That being said, inner mitochondrial membranes cannot be coupled. Please change this in the Abstract section and make this more clear on line 72.

We have made appropriate changes.

2.   This review article would be more beneficial if some sort of figure encompassing CypD´s “master regulator” versus “trouble maker” roles are outlined.

We regret that we had not done this in the original and have added Figure 1.

3.   Sentence on line 44,45 and 46 sort of contradicts sentence on line 46 and 47.

We have changed the first sentence to “In addition, how CyPD’s PPIase activity is involved in this process remains unclear, as there are very few reports on the mitochondrial targets of this activity.

4.   On line 48 change “…model of CypD activity in the mitochondrial matrix.” For “…model of known CypD roles in the mitochondrial matrix.”

This change has been made.

5.   Please note on line 57 that complex II is part of the Krebs cycle.

This change has been made.

6.   On line 68, please refer the reader to specific research articles and not only reviews by your group. In other words, change reference #4 for #61. By the way, not everyone has accepted the “Synthasome” concept. For example, note that your Scientific Reports paper (PMID: 29101324) does not show Phosphate Carrier immunodetection alongside ADP/ATP translocase and ATP synthase.

We have added appropriate citations here.

7.   Please cite PMID: 26902508 on line 81. 

This citation has been added here and in reference to additional cyclosporin analogues.

8.   On line 84 cite an excellent review on Hunter and Haworth´s work PMID: 9714722.

This citation has been added here and in reference to signaling via transient PTP opening.

9.   Add a suitable reference on line 98.

References have been added here.

10.On line 85 add a suitable reference after “necrosis”.

A reference has been added.

11.On line 90 add “componentry” or something alike after PTP.

The end of that sentence now states “the molecule or molecules that create, and do not merely regulate, the PTP”

12. Change ubiquinone for ubiquinones on line 112.

This change has been made.

13. The OSCP sentence feels out of context on lines 124-126. Try to merge lines 127-132 in one ATP synthase and CypD paragraph.

We have rewritten this sentence but concentrated on CyPD and ATP synthase, and we have also combined these two paragraphs.

14.By the way, the authors are not citing a key paper showing unchanged PTP responsiveness to calcium and cyclosporin A in OSCP depleted cells (PMID: 28784775).

We have added this reference.

15. Add cyclosporin A (CsA) to the list of abbreviations and update the manuscript accordingly.

This change has been made.

16. Sentence on lines 173-175 is misleading considering that HK and mtCK have not been deleted and PTP under such conditions assessed.

This is an excellent point, and we have replaced “all of these CyPD binding partners” with “ANT, VDAC, and PiC”.

17.For the sake of balance, please cite and explain the findings in PMID: 28289229.

18. On line 180 add a sentence something like “Although these results have been seriously challenged [PMID: 28784775]”

We have added a brief discussion of the data surrounding this controversy, including these and other references

19.On line 186 add a similar sentence and cite PMID: 26581158.

We now cite this paper’s dispute of the SPG7 manuscript.

20. On line 188 change “PTP activity” for “CsA-insensitive, PTP-like activity”.

We have expanded this sentence to more completely describe the electrophysiologic results of these two manuscripts.

21.Please discuss a bit on how can p53 enter the mitochondria on line 205.

We now have a brief discussion of this and added a new reference.

22. Add a citation after “synthasome assembly” on line 249.

This change has been made.

23. On line 254, please note that Molkentin´s group has previously shown that prolyl isomerase-deficient CypD recapitulates the CypD knockout phenotype suggesting CypD enzymatic activity is indeed required for PTP (PMID: 15800627).

This is an excellent point and we now reference it in the paragraph immediately preceding paragraph.

24. On section 8.2, please note that Bos Taurus CypD lacks an equivalent C203 and yet displays a canonical CsA-sensitive PTP.

This has been added. However, we would note that, although there is a general lack of homology in this region of Bos Taurus CyPD compared there are other cysteine residues nearby

25. On line 307 change “…without a mitochondrial…” for “…without an obvious mitochondrial…” and “no phosphatases” for “not known phosphatases”.

This change has been made.

26.On line 331, please mention possible causes for this outcome (check Bernardi´s Letter to the Editor after reference 115).

We have added to the end of the sentence and added this and one other reference.

27.Why don´t you add a figure showing the molecular structure/family of all known drugs targeting CypD.

We have added this as Figure 2.